# Competences Enabling Young Germans to Engage in Activities for Climate Protection and Global Health

**DOI:** 10.3390/ijerph22071111

**Published:** 2025-07-15

**Authors:** Volker Gehrau, Iris Morgenstern, Carola Grunschel, Judith Könemann, Marcus Nührenbörger, Angela Schwering, Christian Fischer

**Affiliations:** 1Department of Communication, Faculty of Educational and Social Sciences, University of Muenster, 48143 Münster, Germany; 2Institute of Educational Science, Faculty of Educational and Social Sciences, University of Muenster, 48143 Münster, Germany; morgenst@uni-muenster.de (I.M.); ch.fischer@uni-muenster.de (C.F.); 3Department of Educational Psychology, Faculty of Psychology/Sport and Exercise Sciences, University of Muenster, 48149 Münster, Germany; carola.grunschel@uni-muenster.de; 4Institut für Religionspädagogik und Pastoraltheologie (Institute for Religious Education and Pastoral Theology), Faculty of Catholic Theology, University of Muenster, 48149 Münster, Germany; j.koenmann@uni-muenster.de; 5Institute of Fundamental and Inclusive Mathematics Education, Faculty of Mathematics and Computer Science, University of Muenster, 48149 Münster, Germany; nuehrenboerger@uni-muenster.de; 6Institute of Geoinformatik, Faculty of Geosciences, University of Muenster, 48149 Münster, Germany

**Keywords:** engagement, climate protection, global health, competence, internet, ethic, religion, mathematic, natural science

## Abstract

The study examines how individual skills influence adolescents’ and young adults’ commitment to climate protection and global health. Based on 21st-century skills and transformative education, it focuses on competences in science, geography, mathematics, religion, ethics, and media. A representative online survey of 783 participants in Germany assessed topic relevance, information-seeking behavior, and active engagement. The results show that subject-specific skills significantly affect engagement, especially when closely related to the topics. Scientific, mathematical, and geographical competences enhance all three engagement dimensions. Media competence primarily increases perceived importance, while religious and ethical competences positively influence all areas. Structural equation modeling reveals a process: perceived relevance leads to information search, which then drives engagement. Thus, competences have both direct and indirect effects on active involvement. The findings highlight that long-term engagement is not only driven by campaigns but also by education that equips young people with the skills to recognize important issues, seek information, and take action independently.

## 1. Introduction

Many young people are actively working to improve the world, from advocating for climate action to volunteering in global health initiatives. At the same time, others acknowledge the importance of these issues but choose not to get involved. This study examines the personal factors that motivate and empower young people to engage in causes like climate action and global health. The focus is on individual competencies, including abilities, skills, and talents, collectively referred to as competence throughout the text. Building on concepts like 21st-century skills, we suggest that core competencies developed in school, along with soft skills such as media literacy, organizational ability, and teamwork, play a crucial role in motivating young people to work toward a better world. Unlike the short-term impact of school projects or health campaigns, our focus is on the long-term influence of competencies such as those in natural sciences, mathematics, geography, ethics, religion, and digital literacy, which inspire and empower young people to take meaningful action for a better future.

### 1.1. Approach

This study is part of an interdisciplinary research project focused on 21st-century skills, transformative education, and actions that contribute to a fulfilling life. The project seeks to identify individual, motivational, infrastructural, and educational factors that inspire young people to engage in causes they find meaningful.

The concept of transformative talent development (see Figure 1) builds on the Transformative Model of Giftedness and Talent Development, which centers the individual while considering their biographical background, as well as their social and educational environments [1]. In this approach, giftedness is seen as the development of an individual’s performance-related potential, shaped by a unique combination of adaptive capability (‘can do’) and transformative personality (‘will do’). Within this model, performance is understood as evolving across various talent domains, encompassing not only individual expertise but also a commitment to the common good, aligned with Sternberg’s Transformational Active Concerned Citizenship and Ethical Leadership (T-ACCEL-Model) [2]. This understanding of transformational giftedness [3] highlights critical (analytical) thinking, creativity, common sense, wisdom, ethics, and passion as key talents for the 21st century [3]. It aims to cultivate social responsibility and empower individuals to actively help shape a sustainable future in response to global challenges [4]. Moreover, learning and educational processes within innovative learning environments are considered key to transforming potential into performance [5]. They play a crucial role in developing competencies as performance potential, which, over time, supports not only the acquisition of subject-specific expertise [6], but also the activation of meaningful engagement. Talent development is seen as an iterative, dynamic, and circular-spiral learning process, where the sustainable growth of competence is closely linked to transformative education [7] within innovative learning environments can, in turn, foster competence for sustainable development. For this purpose, the innovative learning environment is conceived as a multi-level system that mirrors the various levels of action within the educational system, as outlined by Fend [8]. The micro-level of individual actors (e.g., parents, teachers, educators, peers), the meso-level of institutions (e.g., family, kindergarten, school, university), and the macro-level of the system (e.g., educational policy, educational administration) are all considered equally important [9]. This paper focuses on individual competences.

In contemporary geography education, climate protection and global health, two of the most pressing global challenges of our time, are key themes [10]. Geographical methods, such as spatio-temporal analysis and planning, support both the understanding of global dimensions and the development of responses to regional and local impacts. This also includes the critical reflection on geomedia and the interpretation of spatial information in relation to the following: (1) environmental aspects—such as examining global transportation versus the shift toward sustainable mobility in one’s own city; (2) economic aspects—for instance, comparing global supply chains with the localization movement aimed at strengthening local economies; (3) societal aspects—including the creation of more sustainable, resilient, and equitable societies. Research has explored the extent to which education influences individuals’ environmental attitudes and encourages more sustainable travel behavior and transportation choices [11]. Gryl et al. introduced the concept of spatial citizenship, which refers to the ability to critically assess spatial information and actively use geomedia and technology for social participation. Spatial citizenship is becoming increasingly important in geography education as a way to empower students to become critical, digitally literate, and politically engaged citizens. It is considered a key competence for active participation in a democratic society [12]. Transformative education is especially relevant in geography lessons, as it promotes deep learning processes that extend beyond the mere transmission of knowledge about global challenges. It seeks to influence students’ attitudes and behaviors regarding the causes and consequences of these global challenges. Despite widespread discourse on transformative education, there remains a lack of pedagogically grounded models that clearly define how such educational processes can be effectively implemented and evaluated [13]. The extent to which competencies influence engagement and commitment to key challenges such as climate protection and health at global, regional, or local levels remains unclear.

The use of representations to visualize mathematical objects and operations [14] has a long-standing tradition in mathematics education, as do the modeling and mathematical interpretation of data [15,16]. A solid foundation in mathematics, therefore, includes the ability to solve specific problems both independently and collaboratively, to represent and explain relationships and concepts mathematically, and to apply mathematics meaningfully and contextually to phenomena in nature, society, culture, and the world of work [17]. In recent years, the ability to reflect mathematically and critically on data, especially within the context of data science, has gained increasing importance, as young people must navigate a data-driven world to actively shape a future oriented toward the common good [18]. However, there is still limited empirical evidence on how mathematical skills in data analysis and representation contribute to fostering action oriented towards the common good.

In the field of religious education, the topic of ethical learning has gained significant importance in recent years, not least due to ongoing societal developments. In the field of religious education, ethical learning has become increasingly important in recent years, partly in response to ongoing societal developments. For a long time, and in some cases still today, ethical learning within ethics and education has also been addressed under the terms of value formation or value orientation [19]. This also involves examining the ability and skills of young people to act in the interest of the common good, traditionally understood as the bonum commune, or the good of all. Acting for the common good can be understood as engaging with the question of how a good life is possible for everyone, both now and in the future. Relevant competences in this context include ethical judgment, a sense of responsibility, the ability to differentiate between various models of justice, and social skills such as empathy, communication, and reflective thinking [20]. Despite the intensive discussion of this topic in recent religious education discourse, there is still little empirical evidence in this area. In this regard, the empirical findings on the significance of ethics and religion for action oriented towards the common good remain a notable research gap.

Studies in media literacy [21], social media literacy [22], and digital competence [23] has a well-established tradition across various disciplines, such as communication studies, media studies, and media education. On the one hand, such studies measure competencies in using the internet and social media. On the other hand, they encompass a broader range of aspects, including media use, knowledge, skills, ethical considerations, participation, and social pressure, which may closely align with or even overlap the competence areas mentioned above, as well as aspects of engagement [23,24,25]. Various studies have explored the relationship between individual internet competence and engagement in different areas of political and social life. Studies have found evidence of a positive impact of media competence on civic engagement [26,27], social online behavior [28], and political participation [28,29]. In addition, media competence has been incorporated into approaches exploring the pathway from knowledge to action [30,31,32]. In short, internet competence has already been conceptualized as a driver of engagement for a better life.

### 1.2. Engagement for a Better Life

In our model, competences may transform into engagement for a better life. However, it remains challenging to define what ‘engagement for a better life’ truly means and how it can be measured. We adopted a broad approach to capture a wide range of engagement types and areas. Since the target group of our study consists of adolescents and young adults aged 16 to 25, we focused on issues and activities relevant to their concerns and interests. Furthermore, engagement should be understood as a developmental process toward acting for or against something. This developmental process can be modeled in analogy to the Diffusion of Innovation Theory, where individual adoption begins with knowledge and moves through persuasion toward decision-making [33].

If we understand engagement in activities for a better life as a decision to pursue something new, similar to adopting an innovation, then the process begins with acquiring knowledge about an issue and reflecting on its relevance to one’s own actions. In this context, the perceived importance of an issue emerges as a key factor. Only issues that exceed a certain threshold of relevance are likely to be considered, and the likelihood of action increases with the individual’s assessment of that relevance. The influence of perceived issue importance on the attribution of political and social problems is well documented, particularly in the context of second-level agenda-setting effects and the influential role of mass media in shaping perceptions of what constitutes important societal issues [34,35]. As the next step, people need information to form an opinion about the issue and to know about the possibilities of engaging in activities concerning the issue. Such phenomena are often referred to as information seeking or information scanning [36]. Additionally, competence models establish a link between informational needs and social practices [32]. In Diffusion of Innovation Theory, awareness of an innovation, in this case a societal problem, is shaped by information from mass media or one’s social environment, which can foster positive attitudes toward engagement. Nevertheless, a gap often remains between persuasion and actual action [33].

Importance, information seeking, and engagement in actions are all connected to specific issues, which are problems that must be addressed in the pursuit of a better life. However, there is no universally accepted definition of what constitutes such issues. The relevance of issues varies depending on context and on differing societal or political perspectives. Consequently, young people, who are the target group of this study, often prioritize different issues than older generations, reflecting their unique perspectives, life experiences, and concerns within specific social and political contexts. Students—like everyone else—must first be aware of an issue and have a basic understanding of its nature. To ensure this, we drew on items from the 2018 PISA student survey, which assessed students’ familiarity with global issues. Students were asked to rate their familiarity with global challenges, including climate change, global health, migration, international conflicts, hunger, poverty, and gender equality [37]. In our survey, we adapted these issues and asked students to (1) evaluate the importance of each, (2) indicate their information-seeking behavior related to them, and (3) report their engagement in corresponding actions. In the following sections, we analyze the three dimensions of importance, information seeking, and engagement in relation to the issues of climate protection and global health.

### 1.3. Research Gap and Hypotheses

The literature on transformative talent development suggests a link between individual potentials (i.e., adaptive capability, transformative personality, see Figure 1), summarized in this study as competencies, and individual performances like engaging for a better life in certain areas (e.g., climate protection and global health). Competencies have been examined in different domains, taking their relation to common well-oriented engagement into account. However, there is a lack of knowledge on how different domains of competencies are related to engagement, and especially how they work together. In addition, engagement includes different levels. We do not yet know which competence is important at which level of engagement. Finally, it is still unclear how the different levels of engagement develop and result in individual behavior aimed at a better life. This study addresses these research gaps with a research question and two hypotheses:

Research Question: How do various competencies influence young Germans’ engagement in climate protection and global health initiatives?

**Hypothesis** **1.**
*An increase in competencies leads to greater engagement of young Germans in climate protection and global health.*


Building on our engagement model, we propose a sequential relationship between different levels of engagement:

**Hypothesis** **2.**
*A stronger sense of issue importance increases information-seeking behavior, which, in turn, enhances participation in climate protection and global health movements.*


## 2. Materials and Methods

### 2.1. Survey

This study was part of a larger interdisciplinary research project on 21st-century skills, transformative education, and civic engagement with a focus on the common good. As part of this project, a survey was conducted to assess personal and psychological factors, as well as various competencies developed at school and during leisure time.

The survey was conducted online using soscisurvey.de, a professional platform for online surveys in Germany. It began with an explanation of the survey’s purpose and information on data protection. Participants could proceed only after providing informed consent. All data reported in this paper are based on responses to statements rated on a seven-point scale, with the option to indicate unwillingness to respond or to skip questions entirely. As a result, the number of valid responses varied across items. To minimize nonresponse, the questionnaire was thoroughly pretested, streamlined to include only essential items, and reviewed by an ethics board. Detailed logging of the survey process revealed no significant issues. On average, participants completed the survey in 10 min, and no items showed high rates of nonresponse or survey abandonment.

### 2.2. Measures

The criterion variables in this study focused on two key aspects of contributing to a better life: climate protection and global health. Each aspect was evaluated across three levels of engagement.

The first level assessed whether individuals perceived climate protection and global health as important societal issues. The second level examined whether they actively sought information about climate protection and global health. The third level measured their involvement in activities supporting climate protection and global health, such as signing petitions, attending demonstrations, or participating in voluntary work. In total, we analyzed six criterion variables: importance, information seeking, and engagement for each of the two issues, with each variable measured using a single item.

The predictor variables focused on competencies in geography, natural sciences, mathematics, religious education, as well as media and internet use. While these competencies were selected based on the research team’s disciplinary backgrounds, they still represent a broad range of skills relevant both within the school context and for organizing social engagement outside of school. We aimed to use the same items for both 16-year-old adolescents still in school and 25-year-old adults who had left school several years earlier. Additionally, the items reflect the core of each competency area. To identify the most suitable items, up to eight items were initially surveyed for each area of interest. Principal component and reliability analyses were conducted to identify the three or four items that best captured the core of each area of interest while also forming reliable scales. To adequately capture competencies related to internet and social media use, four items were necessary to achieve a reliable scale. For all other areas, three-item solutions were sufficient to meet the reliability criteria.

### 2.3. Sample

The sample was recruited via an online panel in Germany provided by Respondi/Bilendi, Cologne, Germany. Participants were required to be between 16 and 25 years old. A quota sampling method was applied to approximate the German population in this age group with respect to gender and educational background—targeting roughly equal distributions of male and female participants, as well as upper and lower secondary education levels. A total of 972 individuals began the survey. The sample was subsequently reduced for several reasons: speeders (those who completed the survey unusually quickly) were excluded; some respondents were slightly outside the specified age range due to recruitment based on year of birth; and a few late participants were removed to meet quota requirements. The final sample consisted of 783 participants. Of these, 51% had a lower level of education, and 52% identified as female. The average age was 21.8 years (SD = 2.9), with a median age of 22.

### 2.4. Statistical Analysis

Descriptive statistics were calculated for the engagement indicators. Based on substantive validity considerations and exploratory reliability analyses, competence items were preselected, followed by a confirmatory factor analysis (CFA) to validate the competence constructs using structural equation modeling (SEM). Subsequently, SEM was applied in two steps: first, to examine the relationship between competence dimensions and engagement indicators; and second, to model the engagement process through mediation analysis [38]. All SEM analyses were conducted using the lavaan package [39] within the RStudio 2025.05.1 environment. Model fit was evaluated according to the following criteria: a Chi-square/df ratio below 5 indicated an acceptable fit, and below 3 indicated a good fit. The Root Mean Square Error of Approximation (RMSEA) was considered acceptable if below 0.10 and good if below 0.07. The Comparative Fit Index (CFI) was deemed acceptable if above 0.90 and good if above 0.95 [40,41]. Due to the model’s complexity and the high number of degrees of freedom, we did not apply the non-significance criterion of the Chi-square statistic [40]. To interpret the strength of SEM coefficients, we followed Cohen’s classification [42]. Small effects ranged from 0.1 to 0.3, medium effects from 0.3 to 0.5, and large effects began at 0.5.

## 3. Results

The engagement levels, including issue importance, information-seeking behavior, and engagement in corresponding actions, were each measured with a single item on a 7-point scale in the domains of climate protection and public health. Among these dimensions, issue importance showed the highest mean values. The means for information seeking were approximately one scale point lower, and the means for engagement in actions were about one scale point lower again. Hence, young Germans attributed high relevance and importance to the issues of climate protection and global health. However, only part of this perceived importance was reflected in active information seeking, and only a portion of that, in turn, translated into actual engagement. This pattern may indicate an engagement process that begins with perceiving an issue as important, followed by information seeking, and, ultimately, though not always, leading to action. However, while the differences between the mean values do provide support, they are not sufficient to draw such a far-reaching conclusion. Moreover, there was a slight difference between the two issues. On the level of issue importance, both climate protection and global health received a mean value of 5.3. For information seeking and actual engagement, the means were higher for climate protection (4.2 for information seeking and 3.4 for engagement) than for global health (3.9 for information seeking and 3.1 for engagement). The standard deviations revealed substantial individual differences among the young Germans (see Table 1).

To test the usability of the selected items for competencies, a confirmatory factor analysis (CFA) was conducted, with latent variables representing the six areas of competence and determining the three or four related items. Our confirmatory factor analysis met the requirements, with a Chi-square/DF ratio of 2.9, an RMSEA of 0.05, and a CFI of 0.96. Additionally, all items were strongly determined by their corresponding factors, with factor loadings ranging from 0.6 (minimum) to 0.92 (maximum). The loadings for the internet factor were lower compared to the other factors but were stabilized by using four items. As a result, we accepted the factors as latent variables representing competence across the six areas (see Table 2).

The model indicated that the latent variables were all positively interrelated, meaning that higher latent competencies in one area predicted higher competencies in other areas as well. The correlation between competencies in internet use and religion was small. Six correlations indicated medium effect sizes, and seven correlations exceeded the threshold for strong effects. The strongest interrelation was found between competencies in geography and natural science, with a correlation of 0.85 (see Table 2).

Structural equation modeling (SEM) was used to estimate the impact of competencies on issue importance, information seeking, and actual engagement in climate protection and global health. In the first step, a separate model was calculated for each level of engagement, namely, issue importance, information seeking, and actual engagement. All three models met the fit criteria, with RMSEA = 0.05 and CFI = 0.96 for all models, and a Chi-square/DF ratio of 2.6 in the issue importance model, 2.7 in the information seeking model, and 2.9 in the actual engagement model. All the models were effective in predicting the levels of engagement. The explained variance (R^2^) ranged from 20% for the issue importance of global health to 29% for the search for information about climate protection. In addition, each model revealed high correlations between the engagement levels across the different issues. The issue importance of climate protection and the issue importance of global health correlated at 0.48. The search for information about climate protection and the search for information about global health correlated at 0.55. Both correlation coefficients are on the threshold of strong effects. A very strong association was found between the actual engagement for climate protection and the actual engagement for global health (see Table 3).

The models explaining the levels of engagement did not show an overarching structure but instead revealed single or multiple effects of all six competence areas on different aspects of engagement.

Competencies in mathematics were related to the search for information about global health and to the actual engagement in global health. This could be attributed to the COVID-19 pandemic, which was frequently analyzed using ratios of infection or mortality and the probability of severe disease outcomes.Competencies in natural sciences were associated with actual engagement in global health by elevating the perceived importance of the global health issue. Knowledge in biology and chemistry may have been valuable for understanding the dangers of global health crises and the importance of addressing global health issues.Competencies in geography were related to the probability of seeking information on climate protection. Geographic knowledge might even have been a precondition for understanding information about climate.Ethical competencies appeared to be crucial for evaluating the importance of issues, particularly in the case of climate protection, and to a lesser extent, global health. Ethical competence was associated with searching for information about climate protection, while religious competence enhanced the search for global health information and fostered actual engagement in both climate protection and global health. Thus, we found a positive influence on all engagement indicators from either ethical or religious competence. The only significant negative relationship was observed between religious competence and the perceived importance of climate protection, which may well have resulted from multicollinearity.Competencies in internet use was related to the perceived importance of climate protection and, especially, global health. This could be seen as a form of agenda-setting effect, where issues frequently highlighted by the mass media are perceived as more important by the public, compared to those that rarely appear in the media [34]. Due to the COVID-19 pandemic and the Fridays for Future movement in Europe, young people with high internet competence may have been more frequently exposed to information about these issues on the internet or social media, making them more susceptible to agenda-setting effects (see Table 3).

In summary, all correlations between the six competence areas and the search for information, as well as the actual engagement, supported the first hypothesis: higher competencies were associated with greater engagement. However, the correlations with issue importance showed a few negative relationships, with only one being significant, compared to five significant positive correlations.

On a second step, two SEM were estimated to better understand the process of engagement for climate protection and global health. Once again, the six competences were used as predictors for the level of engagement. In addition, importance was used as a further predictor for information seeking, and importance and information seeking as further predictors of engagement.

These models provided a better understanding of the process leading to actual engagement. If the competencies predicted the levels of actual engagement as presented in the SEM above, without one level of engagement predicting the others, we would interpret the result as counterevidence for a systematic process of engagement. In that case, issue importance, information seeking, and engagement in activities related to the issue would be considered independent phenomena. In contrast, if the levels of engagement served as predictors for each other, and the explained variance of one level increased when considering a previous level as a predictor, we would interpret the model as an indication of a developmental engagement process, starting with issue importance, followed by information seeking, and ultimately leading to engagement in activities. In this case, we could examine the indirect effect of competencies on actual engagement, mediated by issue importance or information seeking. The effect sizes of the competence areas predicting the levels of engagement should be greater in the simple models compared to the process models.

According to the fit indices, both models met the criteria, with a Chi-square/DF ratio of 2.8, an RMSEA of 0.05, and a CFI of 0.96. The explained variance of the levels of engagement increased by integrating the other levels of engagement. For information seeking, the variance explained for climate protection rose from 29% to 35%, and for global health, it increased from 28% to 32%. The variance explained for actual engagement increased from 21% to 37% for climate protection and from 27% to 43% for global health. Thus, the additional consideration of individual issue importance improved the explained variance of individual information seeking by about 5 percentage points. Taking issue importance and information seeking into account increased the explained variance of actual engagement by about 15 percentage points. We interpret this result as an indication that individual engagement in actions for climate protection or global health may be better understood as a process starting with issue importance and information seeking, rather than relying solely on the single predictors of actual engagement (see Table 4).

This assumption was also supported by the standardized coefficients. In both models, issue importance had a small but consistently positive effect on information seeking (0.29 for climate protection; 0.2 for global health), while information seeking had a strong positive effect on actual engagement (0.46 for climate protection; 0.48 for global health). It is noteworthy that for both issues, no direct effect was found between issue importance and actual engagement. As a result, the effect of issue importance on actual engagement was entirely mediated by information seeking. The mediated effect can be estimated as the product of the standardized coefficients. Therefore, we identified an indirect effect of issue importance on actual engagement, mediated by information seeking, of 0.13 for climate protection and 0.1 for global health.

The comparison of the coefficients in the simple versus the process models revealed some interesting insights. Some coefficients were nearly identical in both models. This was the case for the impact of religious competence on actual engagement for climate protection, as well as for mathematics and religious competencies in searching for information on global health. These effects represented direct effects of competence on the corresponding level of engagement. Other coefficients were systematically smaller in the process model compared to the simple model. This was the case for competencies in geography and ethics on seeking climate information, and for competencies in mathematics and religion on actual engagement in global health. In three of these four cases, we observed strong effects of competence on the previous level of engagement, along with a positive effect of the previous level of engagement on the current level. We interpreted this as a partial mediation effect. In these cases, part of the effect was direct, going from competence to the actual level of engagement, while another part of the effect passed through the previous level of engagement, being mediated onto the current level (see Table 4).

Hence, Hypothesis 2 was supported: Actual engagement in climate protection and global health appears to be better understood as a process, with competencies influencing issue importance and information seeking, both of which influence and mediate effects on actual engagement, compared to separate models for each level of engagement. The strong effects between the levels of engagement, as well as the significant increase in the explained variance for information seeking and engagement, may partly be due to a direct impact between the levels, but could also indicate mediated effects from other areas of competence.

## 4. Discussion

The study provided strong evidence that individual competencies play a crucial role in the civic engagement of young people in climate protection and global health. In response to the research question, the findings revealed many significant correlations between competencies and the level of engagement among young Germans. Additionally, our first hypothesis was strongly supported: different competencies positively influenced engagement. Young people with higher levels of competence, or those who perceive themselves as more competent, are more likely to view issues like climate protection and global health as important, seek information about these topics, and actively participate in climate protection and global health initiatives.

Our second hypothesis was also confirmed. Competencies not only increased the perception of climate protection and global health as important issues in German society but also supported young people’s information-seeking behaviors on these topics. Furthermore, these competencies enabled young Germans to actively engage in actions for climate protection and global health. We also identified the expected hierarchy in engagement levels: recognizing the importance of these issues led to more frequent and intense information seeking, which, in turn, increased the likelihood of young Germans participating in movements and actions for climate protection and public health.

In conclusion, competencies across various fields are essential for young people’s engagement in climate protection and global health. However, our findings suggest that the relationship is more complex: different competencies play distinct roles in enabling engagement and driving action. For example, competencies in mathematics and natural sciences are important for actual engagement in global health; however, they do not directly lead to an increase in actual engagement. Instead, they enhance the perceived importance of the issues and, more importantly, support information-seeking behaviors, which, in turn, increase the likelihood of actual engagement in climate protection or global health efforts. In contrast, competencies in geography were particularly useful for information seeking about climate protection, which subsequently increased the likelihood of engaging in actions for climate protection.

More surprisingly, the strong effects of competencies in ethics and religion on engagement in climate protection and global health were evident. It appears that having a strong sense of fairness, justice, and societal well-being, along with a vision for creating a better world, is important for young people. Competence in ethical considerations leads young people to evaluate climate protection and global health as important. In contrast, religious competence appears to decrease the tendency to view climate protection as important. This could be due to multicollinearity, but may also stem from other factors, such as a reluctance to view climate change as caused by human behavior. On the other hand, religious competence strongly supports engagement in actions for both climate protection and global health. Participation in religious activities might also serve as a trigger for engagement in other social issues.

Moreover, competencies in using new media and the internet are crucial for engaging in climate protection and global health issues. This does not necessarily imply that media campaigns directly drive engagement, though that may be the case. Instead, the focus is on the importance of the internet and media literacy in making these issues relevant to young people, which appears to be a necessary precondition for active engagement in these areas.

When evaluating our results, several limitations should be acknowledged. First, our approach to modeling and measuring competences represents an initial step within the framework of a pilot study. We concentrated on school-related competencies, which align closely with our research focus in the context of formal education. However, our scope was limited to a subset of these competencies. Other areas—such as language, history, sports, and the arts—may also significantly contribute to fostering engagement and supporting a fulfilling life. Soft skills like working well with others, staying focused, and being self-motivated are important and should be explored more deeply in future research. Addressing complex global challenges—such as climate change or public health crises —may require not only academic competencies but also those that promote pro-social behavior and effective decision-making under uncertainty. These include less easily measurable capacities such as interpersonal responsiveness, adaptive emotional regulation, and ethical reasoning. Regardless of the individual positions held by members of our research team, the project presented here is grounded in the principles of critical rationalism as formulated by Popper. We are aware that critical experiential dimensions often elude quantification and therefore require qualitative designs. In this respect, our research sheds light on only a small part of the overall picture.

It should also be noted that some of our survey items were not optimally designed. For some of the competence areas, the items were not clearly aligned, and while certain items met statistical criteria, they did not adequately capture the core of the intended competence. This was partly due to relying on item suggestions from other studies that addressed individual competences in a broader, less specific manner. Further research is needed to develop items that are both reliable and closely aligned with clearly defined competence areas. In developing our survey instrument, we did not include items related to fear and concern regarding climate change and global health. Including such items might have provided greater nuance to the dimension of importance. At the level of actual engagement, future research should also take into account relational forms of action, such as (1) specific reactions of adolescents that (2) are private, spiritual, or intimate in nature, and (3) are expressed more through forms of restraint, renunciation, or refusal.

Additionally, modeling a set of competences as predictors within a linear framework presents challenges related to multicollinearity. The more closely two competence areas are interrelated, the more difficult it becomes to estimate their individual influence on a given criterion without bias. In such cases, apparent negative relationships may result from strong positive correlations mediated through other variables. In our results, such effects may have occurred between competence in religion and perceived importance, likely due to strong correlations between both variables and ethical competence. Beyond developing improved items and applying more sophisticated statistical models to assess the impact of a broader set of competences on the process of engagement across various fields, additional methods should be used to enrich and validate the findings. One promising approach could involve conducting in-depth interviews with young people to explore their pathways to engagement and the role specific competences play in supporting that process.

## 5. Conclusions

Young people may feel overwhelmed by the complexity and urgency of climate change and global health challenges [43]. Our findings highlight the competencies that schools need to foster in order to help students navigate the potentially overwhelming flood of information. Youth responses to climate change are strongly shaped by complex and multifaceted emotions such as anger, overwhelm, denial, and moral injury [44]. While emotions were not the primary focus of our study, they may help explain a portion of the remaining unexplained variance, which ranges between 70 and 80 percent. Hence, other influencing factors such as personality, political orientation, or motivation, which were not included in our analysis, may significantly affect engagement in climate protection and public health. However, our data provide evidence that individual competences not only foster interest in these issues but also empower young people to actively participate in related initiatives. This finding is significant, as it shifts the focus from specific activities, campaigns, or movements to the underlying skills that enable and sustain such engagement. Efforts to engage young people in climate protection and global health often center on the issues and related events. In schools, for instance, students learn about the challenges of climate change and the global effects of pandemics. Civil movements and events like Fridays for Future demonstrations offer valuable opportunities for youth engagement. While such concrete actions and information are important, our findings underscore the vital role of fundamental competencies in empowering young people to actively participate in these causes.

To understand the underlying facts and connections related to climate change and global health, young people need a solid foundation in the natural sciences. This does not require mastering every detail, but rather developing awareness of key concepts, the ability to identify problems, and a drive to seek solutions. Our findings indicate that competencies in natural science, mathematics, and geography not only support but may be essential for sparking interest and engagement in tackling these challenges. However, our results go even further.

Even competencies not directly related to climate protection or global health are essential for helping young people recognize the importance of these issues and take meaningful action. A strong moral compass is especially critical. Ethical and religious literacy supports both the search for information and active engagement. To get involved, young people must be able to evaluate which problems matter most and which solutions are effective, fair, and just.

In summary, we advocate for a long-term strategy that prioritizes the development of fundamental competencies, skills, and abilities to empower young people to actively shape a better future. One key finding is the importance of religious knowledge and ethical competencies. Schools must ensure that these topics remain integral to teaching and classroom discussion. In the short term, informing students about current climate and health issues and encouraging participation through campaigns can be effective. However, numerous issues compete for attention and resources, often influenced by interest groups, sometimes without young people being fully aware. From a long-term perspective, it is essential to cultivate core competencies that enable young people to independently identify, understand, and critically evaluate societal challenges. Key areas include science, mathematics, geography, ethics, and religious education, as well as digital literacy for navigating the internet responsibly, alongside potentially many others.

## Figures and Tables

**Figure 1 ijerph-22-01111-f001:**
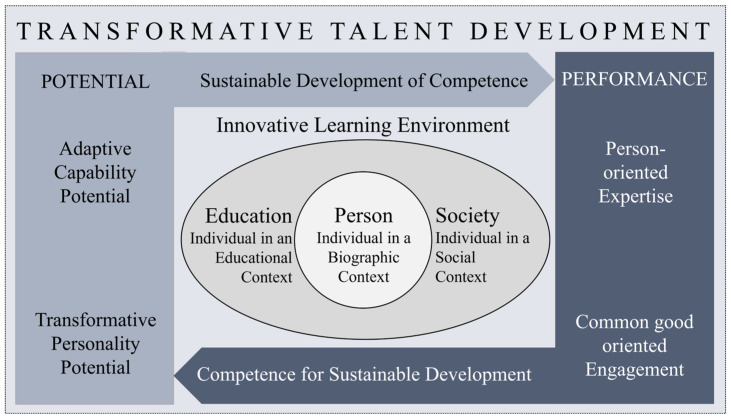
Concept of transformative talent development (own illustration) Performance is depicted in a darker blue to reflect its reliance on potential (light blue), while also indicating the need for additional contributing factors.

**Table 1 ijerph-22-01111-t001:** Engagement criteria for climate and public health.

	Climate		Global Health	
	M	(SD)	M	(SD)
Importance	5.3	(1.6)	5.3	(1.4)
Info seeking	4.2	(2.0)	3.9	(1.9)
Engagement	3.4	(2.1)	3.1	(2.1)

**Table 2 ijerph-22-01111-t002:** CFA of the competence items.

	Loading of Items				Correlation Between Latent Variables				
Latent Variable	Item1	Item2	Item3	Item4	Nat	Geo	Eth	Rel	Int
Mathematics	0.83	0.79	0.83		0.68	0.55	0.35	0.43	0.41
Natural science	0.77	0.84	0.82			0.85	0.49	0.42	0.58
Geography	0.71	0.79	0.79				0.54	0.39	0.59
Ethics	0.81	0.82	0.75					0.45	0.62
Religion	0.92	0.85	0.75						0.20
Internet	0.62	0.60	0.71	0.70					

Chi^2^/DF 394/137 = 2.9; RMSEA = 0.05; CFI = 0.96.

**Table 3 ijerph-22-01111-t003:** SEM of engagement indicators by competences.

	Importance		Info Seeking		Engagement	
	Climate	Global Health	Climate	Global Health	Climate	Global Health
Mathematics	−0.04	−0.13	0.10	0.21 ***	0.11	0.17 ***
Natural science	0.11	0.30 ***	0.20	0.00	0.04	0.02
Geography	0.08	−0.17	0.21 *	0.15	0.17	0.11
Ethics	0.31 ***	0.19 **	0.16 *	0.05	0.04	−0.06
Religion	−0.17 ***	0.00	0.00	0.22 **	0.22 ***	0.36 ***
Internet	0.14 *	0.25 ***	−0.04	0.08	−0.01	0.01
R^2^	0.22	0.20	0.29	0.28	0.21	0.27
Correlation	0.48		0.55		0.73	
Chi^2^/DF	429/163 = 2.6		442/163 = 2.7		474/163 = 2.9	
CFI	0.96		0.96		0.96	
RMSEA	0.05		0.05		0.05	

Note: * *p* < 0.05; ** *p* < 0.01; *** *p* < 0.001.

**Table 4 ijerph-22-01111-t004:** SEM: engagement process by competence.

	Climate			Global Health		
	Importance	Info Seeking	Engagement	Importance	Info Seeking	Engagement
Mathematics	−0.04	0.11 *	0.07	−0.13 *	0.24 ***	0.07
Natural science	0.10	0.17	−0.06	0.31 ***	−0.06	0.02
Geography	0.08	0.18 *	0.08	−0.17	0.18	0.04
Ethics	0.31 ***	0.06	−0.05	0.19 ***	0.01	−0.08
Religion	−0.17 ***	0.05	0.23 ***	0.00	0.22 ***	0.26 ***
Internet	0.14 *	−0.08	0.00	0.25 ***	0.03	−0.03
Importance		0.29 ***	0.06		0.20 ***	0.00
Info seeking			0.46 ***			0.48 ***
R^2^	0.22	0.35	0.37	0.20	0.32	0.43
Chi^2^/DF	489/176 = 2.8			488/176 = 2.8		
CFI	0.96			0.96		
RMSEA	0.05			0.05		

Note: * *p* < 0.05; *** *p* < 0.001.

## Data Availability

Data are contained within this article.

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
