# Peer review of "Competences Enabling Young Germans to Engage in Activities for Climate Protection and Global Health"

_ijerph, 2025, doi:10.3390/ijerph22071111_

Round 1
Reviewer 1 Report
Comments and Suggestions for Authors
This manuscript offers a timely and valuable contribution to the field of educational psychology and civic engagement by empirically examining the relationship between domain/subject specific competences and youth commitment to climate and global health. The use of structural equation modeling to trace the pathways from competence to perceived relevance to information seeking to engagement provides a sound empirical arc. The manuscript does stay aligned, however, with the developmentalist logic of modern education. It emerges from the hypothesis that knowledge acquisition leads to rational behavior. This framing may limit the deeper transformative potential of the research. I offer the following suggestions to deepen its conceptual and holistic integrity.
The first is to reconsider the framing of competence. While the focus on subject-specific skills is methodologically sound, the framing of competence remains largely cognitive/behavioural. I invite the authors to reflect on how we might re-conceptualize competence as the development of core psychosocial/psychospiritual and moral/ethical competencies including interpersonal responsiveness, adaptive emotional awareness/regulation, and ethical reasoning (in addition to knowledge and skills). These competencies support prosocial engagement and decision-making in complex, shifting contexts. By framing engagement as an outcome of individual perception and action, the study adopts a methodological approach consistent with dominant psychological and behavioural models. Yet this framing may also reproduce neoliberal notions of agency and responsibility. These kinds of framings can often obscure forms of engagement rooted in relational, collective, and systemic entanglements and complicities (Andreotti, 2021). These dimensions are less easily quantified but no less central to transformative responses.
I wonder if your model might overlook more relational dimensions of action that include care, forms of resistance (including refusal to participate in certain actions), and even surrendering to feelings of deep grief. These are arguably important actions in the context of the climate crisis. I also wonder if this model can account for expressions of youth engagement that are more intentionally intimate/quiet/introverted, localized, or even psycho-spiritually rooted (for example, making offerings to the water, praying to the trees). Additionally, is it possible that many youth do not lack knowledge, but are overwhelmed by it? In this instance, your focus on competencies is helpful, as it provides strong evidence and insight about how we might support young people in moving through that overwhelm and support their agency. These other dimensions I mentioned, though less amenable to conventional quantification, are essential for understanding how agency to address issues like climate change is co-constituted within ecological, historical and social contexts (see Verlie, 2022). To address this limitation, future research might adopt mixed-methods or participatory designs that integrate qualitative insights into lived experiences, collective meaning-making, and structural dynamics, thereby expanding the epistemic range of what counts as engagement.
I also want to slightly “trouble” the kind of direct pathway from skill to action that this paper presumes. Emerging research (Hickman et al., 2021; Vamvalis, 2023; Veijonaho, 2023) shows that youth responses to climate change are often shaped my multifaceted and complex emotions including anger, overwhelm, denial, and moral injury. Perhaps a brief discussion about the affective hindrances to engagement including despair and collapse-awareness is warranted in light of this growing evidence base? I felt that the study sealed itself off from these growing affective/psychological realities and that this absence was notable in limiting some of the more transformative potentials of the research and its implications.
One of the most interesting findings in the study is that ethical and religious competencies are significant across engagement dimensions. Yet I found myself wanting to know more about this, or seeking a deeper analysis from the authors. I wondered about the role of meaning, purpose, and hope that may be embedded here. In the face of existential threats, are these competences ones which support the kind of psycho-social/psycho-spiritual resilience that is required to carry the heavy weight of “climate coloniality” (Sultana, 2022)? I was thinking of Panu Pihkala’s extensive research and writing in this area as well.
I wondered if a short methodological reflection on the positionality of the research team, the limits of survey instruments, or the assumptions embedded in structural equation modeling could elevate the integrity of the work. Acknowledging that critical dimensions of experience (as it relates to taking agentic action) can be resistant to quantification would strengthen the credibility of the piece.
Given all that is at stake at this time, might the authors be a bit bolder in the findings in terms of what they might be requiring from education and social systems? I wanted the pedagogical and policy significance of these findings to be more clearly communicated, and for some of the educational/social possibilities being pointed to by this research to be more explicitly articulated as well.
This manuscript offers a valuable and timely contribution to the field and should be considered for publication following some minor revisions that more robustly engage with the complexities of climate response. By deepening its analysis and clarifying the implications for practice and research, the piece has the potential to meaningfully advance the discourse on youth agency, education, and planetary care in a time of profound ecological and social transformation. I thank the authors for their work. I hope these invitations are received in the spirit of collaborative thinking and shared commitment to deeper inquiry in troubling times.
Author Response
|
1. Summary |
|
|
|
Thank you for your invaluable recommendations! I’ve begun reading several of the books and articles you suggested, which have truly given me a new perspective that will significantly shape not only this article but also our future work.
|
||
|
3. Point-by-point response to Comments and Suggestions for Authors
|
||
|
Comment 1: I invite the authors to reflect on how we might re-conceptualize competence as the development of core psychosocial/psychospiritual and moral/ethical competencies including interpersonal responsiveness, adaptive emotional awareness/regulation, and ethical reasoning (in addition to knowledge and skills). These competencies support prosocial engagement and decision-making in complex, shifting contexts. |
||
|
Response 1: We focused on the investigation of school-based competencies, as these lie at the center of our research interest in the context of school education. However, it is important to recognize that addressing complex challenges such as climate change and global health crises may require a broader set of competencies - often difficult to quantify - that support prosocial engagement and sound decision-making in dynamic, uncertain contexts. Such competencies include interpersonal responsiveness, adaptive emotional regulation, and ethical reasoning in addition to ethical knowledge. See revised section Conclusion, line 497-511
|
||
|
Comment 2: By framing engagement as an outcome of individual perception and action, the study adopts a methodological approach consistent with dominant psychological and behavioural models. Yet this framing may also reproduce neoliberal notions of agency and responsibility. These kinds of framings can often obscure forms of engagement rooted in relational, collective, and systemic entanglements and complicities (Andreotti, 2021). These dimensions are less easily quantified but no less central to transformative responses. |
||
|
Response 2 Our study conceptualizes engagement as the result of individual perception and action, thereby reinforcing ideas of personal agency and responsibility. We proceed from the assumption that individuals are autonomous agents who can—and should—take responsibility for mitigating and reversing climate change. Implicit in our model is the belief that the actions we included in our engagement variable are productive and desirable, such that the more of these actions are taken, the better the world becomes. We recognize, however, that this framing may (1) obscure or even overlook other, potentially more relational, collective, or systemic forms of engagement; (2) fail to consider that, in some cases, doing nothing might be a meaningful or appropriate response; and (3) ignore the fact that the very actions we promote also consume resources. We cannot resolve this tension, as we ourselves are deeply embedded in the cultural and epistemological frameworks of our society and academic tradition. Nevertheless, we believe it is both possible and necessary to acknowledge these contradictions and remain critically aware of them. (Our target group, like us, is shaped by our cultural and societal context. Therefore, if we're examining how school-related competencies influence engagement, we must measure engagement as it is understood within our society.) We did not include these considerations in the article due to space constraints but will take them into account in our future work.
|
||
|
Comment 3: I wonder if your model might overlook more relational dimensions of action that include care, forms of resistance (including refusal to participate in certain actions), and even surrendering to feelings of deep grief. These are arguably important actions in the context of the climate crisis. I also wonder if this model can account for expressions of youth engagement that are more intentionally intimate/quiet/introverted, localized, or even psycho-spiritually rooted (for example, making offerings to the water, praying to the trees). |
||
|
Response 3: We assessed a specific, one-dimensional form of action. Next time, we could also consider actions such as the decision to forgo certain activities - like flying - as a form of engagement. An exploratory interview study on youth-specific forms of engagement, including voluntary abstention, could provide valuable insights. This also opens up perspectives for further research. We did not include these considerations in the article due to space constraints but will take them into account in our future work.
|
||
|
Comment 4: Additionally, is it possible that many youth do not lack knowledge, but are overwhelmed by it? In this instance, your focus on competencies is helpful, as it provides strong evidence and insight about how we might support young people in moving through that overwhelm and support their agency. |
||
|
Response 4: Young people may feel overwhelmed by the complexity and urgency of climate change and global health challenges. Our findings highlight the competencies that schools need to foster in order to help students navigate the potentially overwhelming flood of information. See revised section Conclusion, line 537-553.
|
||
|
Comment 5: I also want to slightly “trouble” the kind of direct pathway from skill to action that this paper presumes. Emerging research (Hickman et al., 2021; Vamvalis, 2023; Veijonaho, 2023) shows that youth responses to climate change are often shaped by multifaceted and complex emotions including anger, overwhelm, denial, and moral injury. Perhaps a brief discussion about the affective hindrances to engagement including despair and collapse-awareness is warranted in light of this growing evidence base? I felt that the study sealed itself off from these growing affective/psychological realities and that this absence was notable in limiting some of the more transformative potentials of the research and its implications. |
||
|
Response 5: Youth responses to climate change are strongly shaped by complex and multifaceted emotions such as anger, overwhelm, denial, and moral injury. While emotions were not the primary focus of our study, they may help explain a portion of the remaining unexplained variance, which ranges between 70 and 80 percent. See revised section Conclusion, line 537-553.
|
||
|
Comment 6: One of the most interesting findings in the study is that ethical and religious competencies are significant across engagement dimensions. Yet I found myself wanting to know more about this, or seeking a deeper analysis from the authors. I wondered about the role of meaning, purpose, and hope that may be embedded here. In the face of existential threats, are these competences ones which support the kind of psycho-social/psycho-spiritual resilience that is required to carry the heavy weight of “climate coloniality” (Sultana, 2022)? I was thinking of Panu Pihkala’s extensive research and writing in this area as well. |
||
|
Response 6: Further research is needed to understand why and how religious and ethical competencies influence the different dimensions of engagement. In the face of existential threats, do these competencies foster the kind of psychosocial and psycho-spiritual resilience needed to confront climate anxiety or to bear the burden of “climate coloniality” (Sultana, 2022)? When it comes to concrete action, the social embeddedness of individuals in religious communities may also play a significant role. We did not include these considerations in the article due to space constraints but will take them into account in our future work.
|
||
|
Comment 7: I wondered if a short methodological reflection on the positionality of the research team, the limits of survey instruments, or the assumptions embedded in structural equation modeling could elevate the integrity of the work. Acknowledging that critical dimensions of experience (as it relates to taking agentic action) can be resistant to quantification would strengthen the credibility of the piece. |
||
|
Response 7: Regardless of the individual positions held by members of our research team, the project presented here is grounded in the principles of critical rationalism. We are aware that critical experiential dimensions often elude quantification. In this respect, our research sheds light on only a small part of the overall picture. See revised section Conclusion, line 528-531.
|
||
|
Comment 8: I wondered if a short methodological reflection on the positionality of the research team, the limits of survey instruments, or the assumptions embedded in structural equation modeling could elevate the integrity of the work. Acknowledging that critical dimensions of experience (as it relates to taking agentic action) can be resistant to quantification would strengthen the credibility of the piece. |
||
|
Response 8: Regardless of the individual positions held by members of our research team, the project presented here is grounded in the principles of critical rationalism. We are aware that critical experiential dimensions often elude quantification. In this respect, our research sheds light on only a small part of the overall picture. See revised section Conclusion, line 509-511.
|
||
|
Comment 9: Given all that is at stake at this time, might the authors be a bit bolder in the findings in terms of what they might be requiring from education and social systems? I wanted the pedagogical and policy significance of these findings to be more clearly communicated, and for some of the educational/social possibilities being pointed to by this research to be more explicitly articulated as well. |
||
|
Response 9: Detailed recommendations for curricula or the structure of our school system go beyond the scope of this article. The research group is planning to publish a position paper that will address the educational policy implications derived from the study’s findings. Nevertheless, we included the following statement: One key finding is the importance of religious knowledge and ethical competencies. Schools must ensure that these topics remain integral to teaching and classroom discussion. See revised section Conclusion, line 569. |
||
Reviewer 2 Report
Comments and Suggestions for Authors
Dear Authors
this research is quite interesting, but it needs further improvement for achieving publication.
My main concern is your materials and methods section. There is indication of bias in what you describe:
a) How sure you are about the age of the respondents? Could one 40year old just select that she/he is within the preferred age range? (16-25)
b) The choice to cut respondents in order to achieve balance of genders is quite awkward. You should at least present results of balanced and non-balanced samples. Please better explain why you need the balance, and what you should have done earlier in order not to have this problem (if it is a problem)
c) You perform many analyses without presenting them clearly in the methodology section. For instance, PCA is a significant analysis that needs to be presented in detail, with some citations surrounding it.
d) Moreover, you use SEM analysis in the results section without even mentioning it in the methodology section. This is also true for other statistical measures you use in the results section.
e) Finally, concerning the sample, was there an official sampling method, or anybody could answer the questionnaire? You should clearly mention that. If there is no sampling method used, then it isn't valid to draw conclusions about the population.
Adding to the above, I find that you present two hypotheses without linking them to the previous analysis that you present. I would expect a better link of the introduction and the hypotheses.
Please take the above into consideration and try to improve your paper.
All the best.
Author Response
|
1. Summary |
|
|
|
Thank you very much for reviewing our paper. The remarks on the sample and problems of presenting our methods and results were very helpful and improved our text.
|
||
|
3. Point-by-point response to Comments and Suggestions for Authors
|
||
|
Comment 1: How sure you are about the age of the respondents? Could one 40year old just select that she/he is within the preferred age range? (16-25) |
||
|
Response 1: We are sure as the online panel only invited people for the survey who were in the age group. But the panel provider selected based on the year of birth. We asked the age. Due to the individual birthday, some respondents were not 16 but 15 years old and some 26 instead of 25. Such cases have been excluded. But there were no cases outside the range of 15 to 26 years. See the revised section Sample, line 252-263.
|
||
|
Comment 2: The choice to cut respondents in order to achieve balance of genders is quite awkward. You should at least present results of balanced and non-balanced samples. Please better explain why you need the balance, and what you should have done earlier in order not to have this problem (if it is a problem) |
||
|
Response 2: Thank you for the helpful comment. We believe there is no methodological issue; however, our wording may have been misleading. The study used a quota sample. Unfortunately, the stop procedure did not halt the recruitment process quickly enough. As a result, we had to exclude the last female participants with high levels of education, who would have been automatically excluded if the stop procedure had worked more promptly. See the revised section Sample, line 252-263.
|
||
|
Comment 3: You perform many analyses without presenting them clearly in the methodology section. For instance, PCA is a significant analysis that needs to be presented in detail, with some citations surrounding it. Moreover, you use SEM analysis in the results section without even mentioning it in the methodology section. This is also true for other statistical measures you use in the results section. |
||
|
Response 3: Thank you for the helpful remark. We deleted the information about the statistical procedure from the results section. Instead, we created a new section (2.4 Statistical Analysis), where all relevant information is consolidated to provide a clearer understanding of the analytical procedure. See new section Statistical Analysis, line 265-280.
|
||
|
Comment 4: Finally, concerning the sample, was there an official sampling method, or anybody could answer the questionnaire? You should clearly mention that. If there is no sampling method used, then it isn't valid to draw conclusions about the population. |
||
|
Response 4: It was a quota sample. See our first comments.
|
||
|
Comment 5: Adding to the above, I find that you present two hypotheses without linking them to the previous analysis that you present. I would expect a better link of the introduction and the hypotheses. |
||
|
Response 5: We added a new paragraph summing up the literature, identifying a research gap and deducing a research question and two hypotheses for it. See the new section Research gap and hypotheses, line 190-210. |
||
Reviewer 3 Report
Comments and Suggestions for Authors
Introduction:
-
Some paragraphs are too long; please consider condensing them.
-
The authors need to explicitly state the literature gap that this study aims to address.
-
The novelty of this study needs to be emphasized more clearly.
Materials and Methods:
-
Lines 188–199: Please restructure these sentences into a coherent and easy-to-understand paragraph.
-
What sampling technique was used in this study? Please clarify how the chosen sample size is representative of the study population.
Results:
-
In some paragraphs, the authors include reference citations. Please note that references should not be included in this section.
-
Lines 386–409: The paragraph is too long; please consider breaking it into smaller parts or condensing the content.
Discussion:
-
No reference sources are cited in this section. The authors only describe the findings of the current study without analyzing them in the context of previous research. Please include relevant references from prior studies to strengthen the discussion.
Author Response
|
1. Summary |
|
|
|
Thank you very much for reviewing our paper. The remarks on the sample and problems of presenting our methods and results were very helpful and improved our text.
|
||
|
3. Point-by-point response to Comments and Suggestions for Authors
|
||
|
Comment 1: Some paragraphs are too long; please consider condensing them. |
||
|
Response 1: We tried to shorten long paragraphs in the text.
|
||
|
Comment 2: The authors need to explicitly state the literature gap that this study aims to address. The novelty of this study needs to be emphasized more clearly. Lines 188–199: Please restructure these sentences into a coherent and easy-to-understand paragraph. |
||
|
Response 2: We added a new paragraph summing up the literature, identifying a research gap and deducing a research question and two hypotheses for it. See the new section Research gap and hypotheses, line 190-210.
|
||
|
Comment 3: What sampling technique was used in this study? Please clarify how the chosen sample size is representative of the study population. |
||
|
Response 3: It is a quota sample representative for Germans in the age of 16 to 25 years concerning gender and education level. See the revised section Sample, line 252-263.
|
||
|
Comment 4: In some paragraphs, the authors include reference citations. Please note that references should not be included in this section. |
||
|
Response 4: Thank you for this remark. We deleted all references and methodological information from the result section and included them in a new section 2.4 statistical analysis. See new section Statistical Analysis, line 265-280.
|
||
|
Comment 5: Lines 386–409: The paragraph is too long; please consider breaking it into smaller parts or condensing the content. |
||
|
Response 5: Thank you for this helpful suggestion. We divided the paragraph into two sections addressing (a) the indirect mediated effect and (b) the partially mediated effects See new paragraphs in line 412-435.
|
||
|
Comment 3: No reference sources are cited in this section. The authors only describe the findings of the current study without analyzing them in the context of previous research. Please include relevant references from prior studies to strengthen the discussion. |
||
|
Response 3: Some references from the first review were included here. See Discussion, line 537-541
|
||
Round 2
Reviewer 2 Report
Comments and Suggestions for Authors
Thank you for the changes. The paper is better now. I miss some references at the Statistical Analysis section You present the FA and SEM without any citations. Moreover, for the Figure 1 it should be stated whether is yours, or there is a Source that has to be cited.
The limitations should be moved to the discussion section and not the Conclusions.
You state that the research provided "...a strong evidence...". Since the model explains only 20% to 30% of the variance in the dependent variable, it is important to acknowledge this limitation and explore possible factors contributing to the remaining unexplained variability.
Author Response
(0) Thank you for the changes. The paper is better now.
Thank you for reviewing our paper a second time. We have carefully considered all your comments and addressed each point, as they have been very helpful in improving the quality and clarity of the manuscript.
(1) I miss some references at the Statistical Analysis section You present the FA and SEM without any citations.
We added two references (see lines 672-674):
To clarify the foundation of our mediation analysis for the readers, we have included:
- F. Hayes and T. D. Little, Introduction to mediation, moderation, and conditional process analysis: a regression-based approach, Third edition. in Methodology in the social sciences. New York London: The Guilford Press, 2022.
To disclose to the readers the software and type of programming we used in our analysis, we have included:
- Rosseel, “lavaan : An R Package for Structural Equation Modeling,” J. Stat. Soft., vol. 48, no. 2, 2012, doi: 10.18637/jss.v048.i02.
(2) Moreover, for the Figure 1 it should be stated whether is yours, or there is a Source that has to be cited.
Thank you for pointing out this omission. We have now added 'own illustration' in the footer to clarify the source. (See line 78)
(3) The limitations should be moved to the discussion section and not the Conclusions.
Thank you for this helpful suggestion. We have followed it and relocated the discussion of limitations to the Discussion section. (See line 494-537)
(4) You state that the research provided "...a strong evidence...". Since the model explains only 20% to 30% of the variance in the dependent variable, it is important to acknowledge this limitation and explore possible factors contributing to the remaining unexplained variability.
We appreciated this helpful remark and have addressed it by removing the word 'strong' and adding a brief discussion of other potential influences that were not included in our model: “Hence, other influencing factors such as personality, political orientation, or motivation, which were not included in our analysis, may significantly affect engagement in climate protection and public health.” (See line 544-549)